# Efficient Regeneration of Transgenic Rice from Embryogenic Callus via *Agrobacterium*-Mediated Transformation: A Case Study Using *GFP* and Apple *MdFT1* Genes

**DOI:** 10.3390/plants13192803

**Published:** 2024-10-06

**Authors:** Van Giap Do, Seonae Kim, Nay Myo Win, Soon-Il Kwon, Hunjoong Kweon, Sangjin Yang, Juhyeon Park, Gyungran Do, Youngsuk Lee

**Affiliations:** 1Apple Research Center, National Institute of Horticultural and Herbal Science, Rural Development Administration, Daegu 43100, Republic of Korea; seonaekim@korea.kr (S.K.); naymyowin@korea.kr (N.M.W.); topapple@korea.kr (S.-I.K.); kweonhj@korea.kr (H.K.); yangsangjin@korea.kr (S.Y.); wngus1113@korea.kr (J.P.); 2Postharvest Research Division, National Institute of Horticultural and Herbal Science, Rural Development Administration, Wanju-gun 55365, Republic of Korea; microdo@korea.kr

**Keywords:** genetic transformation, *Agrobacterium*-mediated transformation, embryogenic callus, transgenic rice, *eGFP* reporter gene, *MdFT1* gene, heterologous expression

## Abstract

Genetic transformation is a critical tool for gene manipulation and functional analyses in plants, enabling the exploration of key phenotypes and agronomic traits at the genetic level. While dicotyledonous plants offer various tissues for in vitro culture and transformation, monocotyledonous plants, such as rice, have limited options. This study presents an efficient method for genetically transforming rice (*Oryza sativa* L.) using seed-derived embryogenic calli as explants. Two target genes were utilized to assess regeneration efficiency: green fluorescent protein (*eGFP*) and the apple *FLOWERING LOCUS T* (*FT*)-like gene (*MdFT1*). Antisense *MdFT1* was cloned into a vector controlled by the rice α-amylase 3D (Ramy3D) promoter, while *eGFP* was fused to Cas9 under the Ubi promoter. These vectors were introduced separately into rice embryogenic calli from two Korean cultivars using *Agrobacterium*-mediated transformation. Transgenic seedlings were successfully regenerated via hygromycin selection using an in vitro cultivation system. PCR confirmed stable transgene integration in the transgenic calli and their progeny. Fluorescence microscopy revealed eGFP expression, and antisense *MdFT1*-expressing lines exhibited notable phenotypic changes, including variations in plant height and grain quality. High transformation efficiency and regeneration frequency were achieved for both tested cultivars. This study demonstrated the effective use of seed-derived embryogenic calli for rice transformation, offering a promising approach for developing transgenic plants in monocot species.

## 1. Introduction

Rice (*Oryza sativa* L.) is one of the most important crops worldwide, serving as a staple food for over half of the world’s population, especially in Asia, where it has great cultural significance. Its rich nutritional content, particularly as a key source of carbohydrates, makes it essential in the daily diet of billions [1,2,3]. Economically, rice farming supports millions of people’s livelihoods and plays a crucial role in both local and global markets. Its adaptability to diverse environments, from floodplains to terraced hills, makes it a resilient crop vital for food security, especially in regions prone to climate variability. Advances in rice cultivation, including high-yield and disease-resistant varieties, have further enhanced its importance by helping to meet global food demands and alleviate poverty in rural communities.

Genetic transformation is a fundamental process in biotechnology and genetic engineering, in which the genetic material of an organism is altered by introducing foreign DNA into its genome. This process is facilitated by various methods, such as bacterial transformation, electroporation, microinjection, polyethylene glycol (PEG)-mediated transformation, biolistics (e.g., particle bombardment, gene gun), and *Agrobacterium*-mediated transformation. Each method has specific advantages and limitations based on factors like plant species, cost, and transformation efficiency. Electroporation creates temporary membrane pores for DNA entry, while PEG-mediated transformation uses chemicals for DNA uptake. Both avoid biological carriers. Microinjection precisely delivers DNA but is labor-intensive and unsuitable for large-scale use. Biolistics, which shoots DNA-coated particles into cells, is effective for hard-to-infect species but can cause gene silencing and random DNA integration, disrupting essential genes. Moreover, vectors, such as plasmids, are commonly used to transport foreign DNA into host cells [4,5,6,7]. This technique is crucial for a wide range of applications, from studying gene function to developing genetically modified crops for agriculture and gene therapies in medicine. Genetic transformation is a powerful tool that enables the precise manipulation of genetic material, driving advances in understanding life at the molecular level and offering innovative solutions in various fields.

*Agrobacterium*-mediated transformation is a process in which *Agrobacterium tumefaciens* (or *Agrobacterium rhizogenes*) naturally introduces foreign DNA into plant cells using Ti (tumor-inducing) or Ri (root-inducing) plasmids [8,9,10]. When the bacterium attaches to a wound site in a plant, it activates virulence genes that process and transfer a specific DNA segment, known as T-DNA, from the plasmid to the plant cell. The T-DNA is then integrated into the plant genome, where it expresses foreign genes and confers new traits to the plant [4,11,12,13,14]. This method, which is widely used in genetic engineering, allows the stable incorporation of desired genes into plant genomes, making it a powerful tool for creating genetically modified plants [15]. *Agrobacterium*-mediated transformation is a highly efficient and widely used method for introducing foreign DNA into plant cells. It offers advantages such as targeted gene integration, low transgene copy number, stable and heritable transformations, and minimal tissue damage compared to other methods, such as gene guns or electroporation [16,17,18]. However, *Agrobacterium*-mediated transformation is less effective in monocots compared to in dicots, leading to its limited use in monocots. While *Agrobacterium*-mediated transformation is more commonly used for dicots, it has also become increasingly successful in monocots with the development of modified strains of *Agrobacterium* and optimized protocols for monocot species. *Agrobacterium*-mediated transformation is preferred today due to its more precise gene integration and higher efficiency in many cases.

*Agrobacterium*-mediated transformation has been successfully employed in various plant species, including tobacco, rice, maize, tomato, and soybean. Numerous studies have demonstrated the effectiveness of this method for generating transgenic plants of these species. Tobacco is one of the species most frequently transformed using this technique, leading to various genetic modifications and trait improvements [19,20,21]. Additionally, *Agrobacterium* has been used for rice transformation [22,23,24,25,26], providing a platform for the development of genetically modified varieties with enhanced agronomic traits [27,28]. Similar successes have been reported in maize, where *Agrobacterium*-mediated methods have facilitated the introduction of new traits [29,30]. Tomatoes, soybeans, and potatoes are significant examples of species transformed using this approach, highlighting the versatility and wide applicability of *Agrobacterium*-mediated transformation in plant biotechnology [29,31,32,33,34]. In rice, *Agrobacterium*-mediated transformation has been studied extensively in both indica and japonica varieties, revealing significant differences in transformation efficiency. The japonica variety ‘Nipponbare’ shows higher rates of callus induction and regeneration, making it more responsive to transformation [24]. In contrast, indica varieties tend to have lower transformation efficiency due to poor tissue culture response and genetic recalcitrance [35]. Advances in genome editing technologies, such as CRISPR-Cas9, have further increased the utility of *Agrobacterium* transformation, particularly in japonica rice ‘Kitaake’ [36], while efforts to improve indica rice transformation continue through various innovations [37]. Similar to other japonica varieties, ‘Dongjin’ exhibits high transformation efficiency. Jeon et al. (2000) successfully created a large T-DNA insertion mutant population using ‘Dongjin,’ confirming its effectiveness for genetic studies [38]. Furthermore, due to its strong ability to induce callus formation, ‘Dongjin’ has been widely used for the production of various therapeutic recombinant proteins through transgenic embryogenic cell calli suspension culture systems [39,40,41,42]. However, research on ‘Samkwang’ remains limited, though its japonica background suggests it may have comparable transformation efficiency. In recent studies, we used seed-derived embryogenic calli of ‘Samkwang’ as explants for genetic transformation, successfully generating transgenic rice calli overexpressing the *MdCHS* gene via *Agrobacterium*-mediated transformation [43]. In another study, using the same approach, we not only obtained transgenic calli but also regenerated whole transgenic plants from those callus lines in ‘Samkwang’ [28]. Despite these successes in generating transgenic rice for gene function studies, key factors, such as callus induction rate, transformation efficiency, and regeneration frequency, have not yet been thoroughly investigated in important Korean varieties, particularly ‘Samkwang’. Therefore, additional research is required to assess the efficiency of gene transfer and optimize protocols to further expand the use of Korean rice varieties in genetic research.

Rice, as well as other monocot plants, face challenges associated with genetic transformation, where the use of explants is more limited compared to dicots. In addition, previous methods for transforming rice relied on complex culture media, resulting in time-consuming preparation and increased costs. Therefore, by presenting a simplified procedure using embryogenic calli as explants, this study aimed to streamline the process of obtaining transgenic rice, making it more efficient and cost-effective. We developed an efficient genetic transformation method using *Agrobacterium* to introduce the foreign genes *eGFP* and *MdFT1* into the rice genome. The *eGFP* gene enabled clear visualization of transgene expression, while introducing the antisense *MdFT1* gene effectively modified key agronomic traits, such as flowering time, plant height, and yield potential, highlighting the biological impact of genetic modifications. Finally, transgenic rice plants were established via an in vitro culture system using embryogenic calli as explants. Furthermore, we evaluated the gene expression on the phenotype (agronomic trait) in subsequent generations of transgenic rice plants with heterologous apple *MdFT1* gene expression. This genetic transformation system offers crucial technical tools for functional gene analysis, uncovering molecular mechanisms in key rice traits, and extends the potential for application to other crops.

## 2. Results

### 2.1. Binary Expression Vector for Rice Transformation

To obtain genetic material for rice transformation, two genes of interest (GOI), *eGFP* and *MdFT1*, were cloned into two binary expression vectors driven by different promoter systems. The *eGFP* gene was fused to *Cas9* under the control of the rice ubiquitin promoter (Ubi) with the NOS as the terminator, resulting in the recombinant expression vector Ubi::Cas9–eGFP (Figure 1a and Appendix A). The *MdFT1* gene was cloned into the binary vector in an antisense orientation through homologous recombination, driven by the rice alpha-amylase (Ramy3D) promoter with the 3’-UTR as the terminator creating the Ramy3D::MdFT1 expression vector (Figure 1b). Both binary expression vectors used the same selection marker gene, *HygR*, which confers hygromycin B resistance. These binary expression vectors were separately transformed into *A. tumefaciens* EHA105, which was then used for embryogenic callus transformation.

### 2.2. Establishment of Transgenic Rice Using Agrobacterium-Mediated Transformation via In Vitro Culture System

Using an in vitro tissue culture system, 79 transgenic lines of Ramy3D::MdFT1 and 67 transgenic calli lines of Ubi::Cas9–eGFP were obtained (Figure 2, Appendix A). After in vitro culture in callus induction media (N6CI), seed–derived embryogenic calli were used as explants for genetic transformation (Figure 2a–c). Following *Agrobacterium* infection and co-culture, the infected rice calli were grown on a selective medium containing hygromycin B (N6SE). Hygromycin-resistant calli began to differentiate with the appearance of new calli after 3–4 weeks. In contrast, false or non-transformed calli did not grow on N6SE, turned brown, and gradually died (Figure 2d). Putative transgenic hygromycin-resistant lines were screened and continuously propagated on selection medium (N6SE) (Figure 2e,f). Subcultures were conducted every 3 weeks. For shoot differentiation, transgenic rice calli were transferred to the shooting medium (MSS). Bud differentiation was initiated from the calli after 4–6 weeks (Figure 2g,h) and quickly developed into mature shoots with leaves (Figure 2i,j). Roots began to appear within 1 week of culture and elongated rapidly during root differentiation. Whole seedlings with roots were obtained on the rooting medium (MSR) (Figure 2k). After 4–6 weeks of culture on rooting medium, whole seedlings, including leaves and full roots, were transplanted to the soil and grown in a glass house (Figure 2l). Subsequently, we obtained transgenic rice plants via an in vitro culture system, from seed-derived embryogenic calli as the initial materials to whole seedlings with roots.

Putative transgenic hygromycin-resistant lines were continuously propagated on N6SE via subculturing. The calli of the putative transgenic hygromycin-resistant lines were sampled for screening using gDNA PCR (Figure 3 and Appendix A). To screen transgenic callus lines with the introduced Ram3D::MdFT1 vector, a specific primer set for *MdFT1* amplification (set 3, Table 1), which amplified a 525-bp fragment (Figure 3a), was used. To screen transgenic callus lines expressing Ubi::Cas9–eGFP, a specific primer set for *eGFP* amplification (set 2, Table 1) was used to amplify a 717-bp fragment (Figure 3b). Additionally, the presence of *HygR* selection marker gene in both Ram3D::MdFT1 and Ubi::Cas9–eGFP was confirmed using a pair of primers (set 1, Table 1), which amplified the 563-bp fragment (Figure 3a,b). Moreover, RT–PCR analysis of *MdFT1* mRNA in transgenic calli showed that *MdFT1* was transcribed in rice cells (Figure 3c).

A high transformation efficiency and regeneration frequency were achieved for the two cultivars. As shown in Table 2, the transformation efficiency was up to 53.4 ± 1.6% and 47.4 ± 3.4% for Ubi::Cas9–eGFP (‘Samkwang’ cultivar) and Ramy3D::MdFT1 (‘Dongjin’ cultivar), respectively. Moreover, a high regeneration frequency was achieved for two rice cultivars. Average shoot regeneration rate in ‘Samkwang’ and ‘Dongjin’ were 64.8 ± 8.3% and 58.3 ± 8.1%, respectively (Table 2 and Appendix A).

### 2.3. Stable and Inherited Expression of Transgenes in Transgenic Plants

Stable and inherited expression of transgenes in transgenic plants is crucial for the successful establishment of genetically modified crops. To determine whether the target genes introduced into the rice genome could be stably transmitted to the offspring, gDNA PCR amplification was performed. T1 generation plants were generated from seeds harvested from T0 transgenic plants, which were germinated in soil and cultivated in a glasshouse (Figure 4a,b). Leaves of T1 transgenic plants were sampled for gDNA PCR. For molecular confirmation, 20 lines were sampled from the progeny of Ram3D::MdFT1 and Ubi::Cas9–eGFP plants. Successful integration of the T-DNA region into the rice genome was assessed based on the presence of target genes (*MdTFL1* and *eGFP*) and a selection marker gene (*HygR*). Following the same procedure as that used for the PCR screening of transgenic callus lines, transgenes in the T1 progeny were detected in Ram3D::MdFT1 (Figure 4c) and Ubi::Cas9–eGFP (Figure 4d) plants. This indicated that the transgene was consistently inherited and remained stable across generations. The transgenic plants were grown for seed harvesting and further analysis.

### 2.4. Transgenic Rice Expressed Ubi::Cas9–eGFP

To evaluate whether the target gene *eGFP* was functionally expressed in transgenic rice, the green fluorescence of the eGFP protein was assessed using microscopic imaging. Fluorescence was observed in both transgenic calli and T1 plants, including in different organs (leaves, stems, and roots) (Figure 5). Six callus lines of transgenic Ubi::Cas9–eGFP and wild type ‘Samkwang’ were selected for eGFP fluorescence observation. Green fluorescence was observed under the excitation spectra in the calli of the transgenic rice using an in vivo imaging system (IVIS Lumina III) (Figure 5a and Appendix A). GFP fluorescence was not observed in WT callus lines (Figure 5a and Appendix A). As expected, a non-fluorescent transition signal was observed in the WT seedlings (Figure 5b–d and Appendix A). Similar to the callus lines, eGFP fluorescence was visualized in different organs, including the roots, stems, and leaves of the transgenic seedlings. Green fluorescence of the eGFP protein was observed in all tissues of the Ubi::Cas9–eGFP plants (Figure 5b–d and Appendix A). Moreover, eGFP protein expression was visualized at the cellular scale in transgenic calli (Figure 5a, right panel) and in different organs of T1 seedlings (Figure 5b–d, right panel) using a confocal microscope. Supporting the gDNA PCR data of the transgenic calli (Figure 3) and transgenic T1 plants (Figure 4), this confirms that the *eGFP* gene, after integration into the rice genome, was functionally expressed and inherited by the progeny.

### 2.5. Heterologous Expression of the MdFT1 Gene Resulted in Overall Changes in Agronomic Traits of Transgenic Rice

Upon cultivation, transgenic Ramy3D::MdFT1 rice exhibited distinct phenotypic changes, including branching, leaf angle, internode length, plant height, and grain productivity. Phenotyping of agronomic traits was conducted to assess morphological changes in comparison with the WT ‘Dongjin’ cultivar. Introduction of the antisense *MdFT1* gene was aimed at delaying flowering time (heading date, HD) in transgenic rice by suppressing its function. The HD of the transgenic Ramy3D::MdFT1 and WT plants were counted from the sowing date until the emergence of the first panicle (Figure 6a). Transgenic Ramy3D::MdFT1 rice showed a slight delay in HD compared to the WT, with an average HD of 65.6 d in WT and 67.1 d in Ramy3D::MdFT1 (Figure 6b).

Furthermore, transgenic Ramy3D::MdFT1 rice exhibited greater total height and internode length than the WT rice (Figure 7a–c). This may be because the introduction of *MdFT1* affects cell development. To investigate this, cells in the elongation zone of the internode stem were examined under a light microscope, and their sizes were measured to identify differences (Figure 7d–g). Cell size differed between the transgenic and WT plants. Although the cell widths were similar, the cell length of the transgenic plants was significantly greater than that of the WT.

In addition to the overall phenotypic description above, another important agronomic trait that determines yield potential is seed productive ability. Seed production is affected by the number of grains in every panicle and the weight and size of the grains. In this study, transgenic rice in which the apple *MdFT1* gene was introduced showed a lower seed set rate, with a significantly lower number of grains per panicle (Figure 8a,b). However, the grain size of the transgenic Ramy3D::MdFT1 rice was larger than that of the WT. Although grain size did not show significant differences in width between transgenic and WT rice, the grain length of the transgenic rice was significantly greater (Figure 8c–f). This led to a significant increase in the length-to-width ratio and weight of the rice grains (Figure 8g,h). In addition to assessing the appearance of rice grains, we quantified their nutrient quality by analyzing their free amino acid content. Analysis of the free amino acid content of rice grains offers a more in-depth understanding of their nutritional quality, as amino acids are the building blocks of proteins and play vital roles in various biological processes. This also allows for a more accurate understanding of the rice’s health benefits, flavor profile, and suitability of the rice for various culinary applications. High free amino acid content, especially in essential amino acids, enhances overall nutritional quality, making the rice more valuable from both a health and economic perspective. Rice grains from the WT and transgenic lines were analyzed using an automatic amino acid analyzer, with two replicates for each line (Appendix A). The contents of all analyzed amino acids in the Ramy3D::MdFT1 transgenic line were significantly higher than those in the WT (Table 3).

## 3. Discussion

The basic classical method of *Agrobacterium*-mediated transformation to obtain transgenic rice has been described previously [22,24]. However, it requires an expensive medium with many different ingredients and takes a long time to prepare. Specifically, this process requires the preparation of separate and complex media for plant tissue transformation. The present study demonstrated that liquid N6CI medium could be utilized throughout the entire *Agrobacterium*-mediated plant tissue transformation process. In addition, *Agrobacterium* culture is processed by spreading it on solid LB media, making it easy to collect and directly use for plant tissue infection. In contrast, various studies have grown *Agrobacterium* in YEP liquid media, which require a centrifugation step to remove the culture medium before it can be used for plant transformation. Thus, we simplified the transformation method for embryonic rice calli, reducing time-consuming preparation and costs.

Transgenic rice plants were generated using seed-derived embryogenic calli in conjunction with *Agrobacterium*-mediated transformation. This approach has proven to be a robust and reliable method for introducing and achieving stable expression of foreign genes in rice and has traditionally been a challenging task because of the monocotyledonous nature of the plant. The use of seed-derived embryogenic calli as starting material is particularly advantageous. Embryogenic calli are known for their high regenerative capacity, which makes them ideal candidates for transformation [44,45,46]. Unlike other explant sources that may be less responsive or more prone to soma-clonal variation, seed-derived embryogenic calli can provide a more uniform and consistent response to transformation and regeneration protocols, which is crucial for achieving high transformation efficiency and the subsequent regeneration of healthy, viable, transgenic plants [47]. As a potential explant, transgenic maize has been obtained from embryogenic calli using particle and microprojectile bombardment transformation methods [48,49].

The *Agrobacterium*-mediated transformation method used in the present study further enhanced the efficiency of gene transfer. *A. tumefaciens* is widely recognized for its ability to transfer foreign DNA into plant cells. This method has several advantages over other transformation techniques such as biolistics (gene gun), including lower cost and higher transformation efficiency [24]. Herein, the use of *Agrobacterium* allowed for the successful introduction of binary expression vectors containing the *eGFP* and *MdFT1* genes (Figure 1) into the rice genome, resulting in the generation of hygromycin-resistant calli (Figure 2) that could be regenerated further into transgenic plants (Figure 3 and Figure 4). In addition, using appropriate *Agrobacterium* strains can increase the likelihood of successful integration of the transgene into the plant genome, thereby enhancing transformation efficiency. In the present study, *A. tumefaciens* strain EHA105 is selected for genetic transformation due to its high efficiency, broad host range, compatibility with binary vectors, non-oncogenic nature, and its ability to effectively transform both dicots and difficult monocot species. Comparison of different *A. tumefaciens* strains on transformation efficiency was conducted in dicots. Chetty et al. (2013), evaluated four *A. tumefaciens* strains to compare their genetic transformation rate in tomato. According to the results, the highest transformation rate (65%) was obtained with the GV3101 strain, followed by EHA105 (40%), AGL1 (35%), and MP90 (15%) [50]. Among them, transgenic plants infected with strain MP90 had fewer transgene insertions, indicating that it was the most efficient in the transfer of single transgene insertions. In conclusion, based on a combination of higher transformation efficiency and lower transgene copy number, the study found that EHA105 is optimal for functional genomics and biotechnological applications in tomato. In tobacco, the highest transformation rate (20%) was obtained with the *Agrobacterium* strain LBA4404, followed by EHA105, GV2260, C58C1 and AGL1 [51]. Four *A. tumefaciens* strains AGL-1, C58C1, EHA105 and LBA4404 were tested for their transformation efficiency in *Mortierella alpina*. AGL-1 and EHA105 had the highest transformation efficiency, whereas LBA4404 failed to transform. The differences in transformation efficiency among the strains were determined based on transcriptional levels of genes virulence (*vir*) genes, suggesting that high transcriptional levels of *vir* genes were important for successful transformation [52]. Historically, monocots (such as maize, wheat, and rice) have been difficult to transform using *Agrobacterium*, since they naturally prefer dicot species. However, EHA105, with its enhanced virulence, has proven more effective in monocot transformation protocols, increasing the success rates for these important crops. Prías-Blanco et al. (2022), reported that EHA105 strains were more efficient for switchgrass transformation than LBA4404 strains. In rice, EHA105 and LBA4404 strains have shown a similar transformation efficiency; however, EHA105 strains had a lower transgene copy number [53]. Similarly, infection ability and efficiency of transformation of *A. tumefaciens* strains EHA105 and LBA4404 have been investigated in maize. According to the results, hypervirulent EHA105 was more infectious than LBA4404 [54]. The characteristics make EHA105 a highly versatile and dependable strain for plant genetic engineering.

High transformation efficiency refers to a high rate of success in the introduction and integration into the genome of an organism during genetic transformation. In plant biotechnology, achieving high transformation efficiency is crucial because it ensures that a large proportion of the treated plant cells incorporate the desired genetic material, which can then be regenerated into transgenic plants. High and efficient transformation and regeneration using seed-derived embryogenic calli as explants has been previously reported in rice. Sahoo et al. (2011) developed a protocol for regeneration of transgenic rice using *Agrobacterium*-mediated transformation in four indica rice cultivars (IR64, CSR10, PB1, and Swarna) [37]. The authors achieved high transformation efficiency (up to 46%) and regeneration frequency (up to 59%) from embryogenic calli using *Agrobacterium*-mediated transformation methods. In the present study, we achieved high transformation efficiency (53.4% and 47.4%) and regeneration frequency (64.8% and 58.3%) for both cultivars, Dongjin’ and ‘Samkwang’, respectively (Table 2 and Appendix A). Two Korean rice cultivars, ‘Dongjin’ and ‘Samkwang,’ were utilized as explants for *Agrobacterium*-mediated transformation of two target genes under the control of two different promoter systems. The *eGFP* gene, driven by the rice ubiquitin promoter (Ubi) in the CRISPR/Cas9 vector, is effective for gene-editing in transgenic rice [55,56,57,58]. For genetic transformation in rice, the rice Ubi promoter was more effective than the widely used constitutive promoter CaMV 35S. Using *Agrobacterium*-mediated transformation of two Australian rice cultivars ‘Jarrah’ and ‘Amaroo’, Upadhyaya et al. (2000) found that the Ubi promoter produced 30-fold higher GUS activity than the CaMV35S promoter in transgenic plants [59]. The apple *MdFT1* gene was expressed under the control of the rice α-amylase 3D promoter (Ramy3D), a strong promoter commonly used for producing recombinant proteins in transgenic rice callus suspension cultures [39,40,41,60].

Next, the integration and stable expression of the target genes *eGFP* and *MdFT1* in the transgenic plants were investigated. The *eGFP* gene served as a reporter, allowing easy visualization of transgene expression through fluorescence microscopy. The clear detection of GFP fluorescence in the transformed calli and various tissues of the transgenic plants provided strong evidence of successful gene integration and expression (Figure 5). In addition, we evaluated the functional expression of transgenic rice using *MdFT1*, which allowed the establishment of the desired phenotype since its function is well-known. The apple *MdFT1* gene is an ortholog of the *FT* gene. The *FT* gene is a key regulator of flowering in plants, encoding the florigen protein, which promotes transition from vegetative to reproductive growth. FT is expressed in leaves, and its mobile protein moves to the shoot apical meristem to induce flowering [61]. The *FT* gene is a central element in the genetic network that governs flowering, making it a key target for both basic research and practical agricultural innovations. Overexpression of *FT* induces early flowering, while loss-of-function mutations in *FT* result in delayed flowering [62]. *MdFT1* plays a critical role in regulating flowering and other aspects of reproductive development in apple trees. The potential of *MdFT1* to manipulate flowering time in apple trees, which is important for breeding programs, has been investigated [63]. Highly conserved across species, *FT* orthologs play a similar role in crops like rice and wheat, making it an important target for breeding programs aimed at optimizing flowering time [64]. This suggests that *MdFT1* suppression can delay flowering as a potential floral gene in rice. In the present study, antisense expression of the apple *MdFT1* gene was designed specifically to modulate key agronomic traits, such as flowering time and plant height. The successful expression of this gene, as confirmed by RT–PCR (Figure 3c) and the observed phenotypic changes (Figure 6, Figure 7 and Figure 8), indicated that the genetic modifications were not only stably inherited but also biologically active in influencing plant development [20,28]. The ability to manipulate such traits through targeted genetic engineering holds significant promise for crop improvement as it allows for the fine-tuning of characteristics that are critical for yield and adaptability. This not only validated the effectiveness of the transformation protocol but also demonstrated the functionality of the introduced genes within the rice genome.

In this study, we present a clear step-by-step guide, that not only provides a protocol on plant tissues’ in vitro culture systems but also on how to establish transgenic plants for the functional study of GOI in the field of genetic engineering (Figure 9). We meticulously documented the entire plant transformation process, from the cultivation of seed-derived embryogenic calli to the cloning of target genes into binary expression vectors and the generation of fully transformed plants. This was followed by comprehensive genetic and phenotypic analyses of the transgenic plants to confirm the success of the transformation. The expression of target genes in transgenic rice has been shown to be stable and heritable. Overall, we developed an efficient *Agrobacterium*-mediated genetic transformation method to obtain transgenic rice plants. The development of genetic transformation systems offers critical tools for functional analysis of GOI, exploring molecular mechanisms behind key traits, and creating genetically modified crops with enhanced traits.

There are several factors that influence transformation efficiency in plants, including the method of transformation, the plant species and genotype, and the type of tissue explant used. The *Agrobacterium* strain, binary vector design, and co-cultivation conditions also play critical roles. Additionally, the plant’s regeneration capacity, proper culture media, and control over gene silencing and transgene copy number significantly impact the success of transformation. Therefore, optimizing these factors might be required to improve the overall efficiency of genetic transformation in certain studies.

## 4. Materials and Methods

### 4.1. Gene Cloning and Construction of Plant Transformation Binary Vectors

First-strand cDNA of apple *MdFT1* was synthesized through RT–PCR using mRNA isolated from the follower buds of ‘Fuji’ apple cultivar. To facilitate cloning, a pair of specific PCR primers (primer set 3, Table 1) was designed to amplify the *MdFT1* gene, incorporating KpnI and XbaI restriction enzyme sites at the 5′ end of the forward and reverse primers, respectively. PCR products were analyzed on 1.0% agarose gels, and the target band of these genes was recovered from gels and purified using a MEGAquick-spin™ plus fragment DNA purification kit according to the manufacturer’s instructions (Cat. No: 17290; iNtRON Biotechonology, Seongnam, Republic of Korea). The fragment was then ligated into a pGEM^®^-T Easy vector (Cat. No: A1360; Promega, Madison, WI, USA) using T4 DNA ligase (Cat. No: 2011B; Takara, Kusatsu, Japan) and transformed into *Escherichia coli* DH5α competent cells (Cat. No: 9057; Takara, Kusatsu, Japan). Subsequently, the plasmid DNA was sequenced (Macrogen, Seoul, Republic of Korea). After sequencing, the cloned sequence of *MdFT1* libraries was blasted/aligned to the original sequence obtained from the NCBI database (GenBank Accession No. AB161112.1) using CLC Genomics Workbench 12 (Qiagen, Hilden, Germany). *MdFT1* fragment was released from the pGEM^®^-T Easy vector plasmid by digesting with KpnI–XbaI restriction enzymes. The *MdFT1* gene was then ligated into the binary vector pMYD317 [42], which was digested with the same restriction enzymes (XbaI–KpnI). XbaI was located downstream of the Ramy3D promoter, and KpnI upstream of the 3′ UTR. The *MdFT1* gene was cloned into the binary vector in an antisense orientation by homologous recombination, resulting in the recombinant expression vector Ramy3D::MdFT1.

For vector construction of the *eGFP* reporter gene, the *eGFP* gene was cloned into pRGEB32 (Plasmid #63142; Addgene, Watertown, MA, USA) by GenScript Biotech (Piscataway, NJ, USA). *eGFP* was fused downstream of *Cas9*, driven by the rice ubiquitin promoter (Ubi), with NOS as the terminator, and designated as the Ubi::Cas9–eGFP expression vector.

These binary expression vectors (Ramy3D::MdFT1 and Ubi::Cas9–eGFP) were transformed into *E. coli* DH5α competent cells to produce a large number of plasmid DNA vectors. Plasmid DNA vectors were isolated from transformed *E. coli* DH5α using a fast DNA-spin™ plasmid DNA purification kit (Cat.No: 17013; iNtRON Biotechonology, Seongnam, Republic of Korea). The presence of the ligated target genes *MdFT1* and *eGFP* in the plasmid DNA of the binary expression vectors was confirmed by enzyme digestion using KpnI–BamHI and XbaI–KpnI, respectively.

### 4.2. Agrobacterium Transformation

Each constructed expression vector, Ramy3D::MdFT1 and Ubi::Cas9–eGFP, was transformed into *A. tumefaciens* EHA105 competent cells using the freeze–thaw method [65]. After transformation, *A. tumefaciens EHA105* cells were thoroughly spread on LB agar plates (BD Difco, Ref 244520, Franklin Lakes, NJ, USA) containing 50 μg/mL kanamycin and 100 μg/mL rifampicin. They were cultured in a sharking incubator at 180 rpm, 28 °C in the dark for 2 d. Single colonies were then picked from the selective plates, inoculated in 5 mL liquid LB broth medium (BD Difco, Ref 244620, Franklin Lakes, NJ, USA) supplemented with antibiotics, and incubated at 28 °C, 180 rpm, overnight in the dark. Plasmid DNA vectors in transformed *A. tumefaciens EHA105* cells were isolated and confirmed by PCR and enzymatic digestion. The transformed *Agrobacteria* were then mixed with 30% glycerol and stored in a freezer as stock at –80 °C.

### 4.3. Culture Media and Plant Materials

Culture media were prepared using the assigned components in specific amounts and concentrations, as presented in Table 4. The pH was adjusted to the desired value prior to autoclaving. Media were autoclaved at 121 °C for 10 min, then cooled down to approximately 60–65 °C, followed by adding hormones, antibiotics, and acetosyringone. Note that some components were added at this temperature to prevent the loss of activity due to autoclaving. The media were mixed well, poured onto culture dishes and bottles, and allowed to solidify and dry on a clean bench.

Mature seeds of rice *O. sativa* L. from two Korean rice cultivars, ‘Dongjin’ and ‘Samkwang’ were utilized for in vitro culture. Rice seeds were dehusked to remove the seed cover, and 10 mL of the dehusked seeds were collected in a 50 mL falcon tube for surface sterilization. Briefly, rice seeds were prewashed in 40 mL of sterile water by soaking for 2 min, washed with 70% ethanol for 5 min, and rinsed with a 2.5% sodium hypochlorite (NaOCl) solution containing 0.01% Tween20 (P9416; Sigma Aldrich, Munich, Germany) for 10 min. They were finally rinsed 3–5 times with sterile water to remove any residual disinfectants. During the sterilization process, 40 mL of the disinfectant was used, and the falcon tube was shaken frequently to increase the sterilization efficiency. Sterilized seeds were blotted on sterile 3M Whatman filter paper before being transferred to the culture medium. Rice seeds were cultivated in vitro in various media with different components at specific cultivation stages, as listed in Table 4. Cultivation was conducted in a culture room under controlled conditions: 25 °C, 100 μM.m^−2^.s^−1^ light intensity, and a 16-h light/8-h dark photoperiod.

### 4.4. Rice Embryogenic Callus Induction and Transformation

To prepare the plant explants for genetic transformation, sterilized rice seeds, which were obtained as described in Section 4.3, were cultivated on callus induction medium (N6CI) with 10–15 seeds per plate. After 15 d of cultivation, the rice scutellum-derived calli were cut from the in vitro germinated seeds using a sterile scalpel and tweezers. The detached embryogenic calli were placed in fresh N6CI and cultured for 5 d before being used for genetic transformation.

The *Agrobacterium*-mediated transformation method modified from a previous study [43] was utilized for embryogenic rice calli transformation. Briefly, 10 µL of transformed *Agrobacterium* stock, prepared as described in Section 4.2, was inoculated in 5 mL liquid LB broth medium with 50 μg/mL kanamycin and 100 μg/mL rifampicin and incubated overnight. The next day, 100 µL of overnight cultured agrobacterial cells were thoroughly spread on the LB agar-supplemented antibiotics plate using a cell spreader and incubated overnight at 28 °C in the dark. Subsequently, 2–3 mL of N6CI liquid medium was added to the plate, agrobacterial cells were harvested using a cell scraper, and the cell suspension was collected in a 50 mL falcon tube using a pipette. Harvested cells were diluted in N6CI liquid medium, adjusted to a final OD 600 of 0.5, and then supplemented with 200 µM of acetosyringone.

Seed-derived embryogenic calli of ‘Dongjin’ and ‘Samkwang’ cultivars were used as explants for the transformation of the binary expression vectors Ramy3D::MdFT1 and Ubi::Cas9–eGFP, respectively. Rice embryogenic calli (40–60 calli from four plates) were prepared as described above and immersed in 40 mL of cell suspension for *Agrobacterium* infection. This transformation was conducted in a 50 mL falcon tube for 10 min with gentle shaking to improve agrobacterial cell infection. After *Agrobacterium* infection, the infected rice embryogenic calli were blotted on multiple layers of sterile 3M Whatman filter paper before being transferred to N6CO medium (Table 4) and co-cultured in the dark for 3 d. After co-cultivation, rice calli were prewashed several times with sterile water to remove calli overcovered with *Agrobacterium* and then rinsed once with 500 mg/L cefotaxime in sterile water. Rice calli were then blotted on sterile 3M Whatman filter paper for drying, then placed on selection medium (N6SE). Transgenic hygromycin-resistant calli developed after 3 weeks under antibiotic selection conditions. The cultivation conditions were the same as those described in Section 4.3.

### 4.5. Transgenic Plants’ In Vitro Cultivation

Putative transgenic hygromycin-resistant lines were propagated on N6SE via subculturing. To obtain transgenic plants, calli were transferred to MSS (Table 4) for shoot induction. Mature shoots were then separated and transferred to MSR for rooting. Cultivation was conducted in vitro in a culture room as described in Section 4.3.

Whole seedlings with roots were established and prepared for ex vitro culture (i.e., sown in soil). Before transplantation, culture bottles were gradually opened for 1 d until the cap of the culture bottle was completely opened. Rice seedlings were removed from the culture bottles, rinsed under running tap water to eliminate residual medium, and the upper leaves were trimmed. The roots were wrapped in Kimwipes and placed in 50 mL Falcon tubes filled with water for 2 d of acclimatization. Seedlings were transplanted into the commercially purchased garden soil in growth pots (depth: 24 cm × diameter: 12 cm) and grown in a glass house with the following cultivation conditions: temperature of 32/25 °C (day/night), 60% humidity, and natural light. Rice plants were cultivated in 2021 (for the T0 generation) and 2022 (for the T1 generation) during warm periods from late April to early October, representing the rice-producing season in South Korea.

### 4.6. PCR Analysis

To select transgenic lines, PCR analysis was employed to identify the transgene in the genomic DNA of the transgenic plants. We conducted two rounds of PCR at two different stages of plant generation: the callus and seedling stages. In the first round, putative transgenic lines obtained from hygromycin-resistant selection media were sampled for genomic DNA (gDNA) isolation. Leaves of T1 plants in the vegetative stage (before panicle initiation) were sampled for the second round of gDNA PCR.

The calli and leaves of the transgenic lines were collected in 2.0 mL Eppendorf tubes containing two stainless steel beads (5 mm), pre-frozen in liquid nitrogen, and pulverized using a TissueLyser II (Qiagen, Hilden, Germany). Homogenized tissues were processed for gDNA extraction using a DNeasy^®^ Plant Mini Kit (Cat. No: 69204; Qiagen, Hilden, Germany). PCR was conducted at a total reaction volume of 20 μL, including 1 µL of gDNA, 1 µL of each forward and reverse primer, and 17 µL of DNase-free sterile water using the Maxime™ PCR PreMix (i-StarTaq) (Cat. No: 25167; Intron, Seongnam, Republic of Korea). Integration of the T-DNA region (harboring the target genes) into the rice genome was investigated using specific primer sets (Table 1). To detect the selective marker gene *HypR*, as well as reporter genes *eGFP*, and *MdFT1*, PCR was conducted with primer sets 1, 2, and 3, respectively. PCR was processed as follows: initial denaturation step of 95 °C for 5 min; followed by 32 cycles at 95 °C for 45 s, 58 °C for 45 s, and 72 °C for 1 min; and a final extension step at 72 °C for 5 min. The PCR products of each target gene were separated on 1.0% agarose gels, and the amplicons were visualized using the Image Lab^TM^ program (Bio-Rad Laboratories, Inc., Hercules, CA, USA).

### 4.7. RNA Isolation, cDNA Synthesis, and Quantification of Transgene Expression

To quantify the transcript expression of the *MdFT1* gene in transgenic rice, callus tissues were used for RNA isolation. Total RNA extraction and cDNA synthesis were performed as previously described [20]. A real-time reverse transcription–polymerase chain reaction (RT–PCR) assay was used to quantify *MdFT1* mRNA expression level. The transcript expression of *MdFT1* in transgenic callus tissues was normalized to that of the rice reference gene *OsAct1* (Accession No. KC140129.1). RT–PCR was carried out in 20 μL of mixture reaction containing 1 µL of cDNA, 1 µL of each forward and reverse primer, and distilled water using the Maxime™ PCR PreMix using running conditions, as described in a previous study [28]. The primers used for RT–PCR of *MdFT1* and *OsAct1* are listed in Table 1 as primer sets 3 and 4, respectively. Amplicons were separated on 1.0% agarose gels and visualized using the Image Lab^TM^ program.

### 4.8. Plant Phenotyping

The T1 generation of Ramy3D::MdFT1 rice cultivated in 2022 was used for plant phenotyping, as previously described [28]. The heading date was recorded when the first panicle emerged. Leaf angle was measured during the vegetative phase for the third and fourth leaves using a protractor. Branching (tiler number) was evaluated during the vegetative phase. Plant height and internode length were measured during the ripening phase, and plant height was determined from the soil base to the panicles. Number of grains per panicle, grain weight, and grain size were assessed at maturity.

### 4.9. Microscopic Imaging Analysis

A light microscope (Axioskop 2; Carl Zeiss, Oberkochen, Germany) was used to analyze stem elongation at the cellular level. Sample preparation and light microscopy were performed as described previously [20]. Cell size was measured using Olympus cellSens Standard software (Olympus, Shinjuku, Tokyo, Japan).

To observe GFP fluorescence in calli and whole plants, including in different organs (leaves, stems, and roots), an In Vivo Imaging System (IVIS Lumina III; PerkinElmer, Hopkinton, MA, USA) was employed. The specimens for observation were pre-adapted to complete darkness in a dark chamber for 1.5 h prior to evaluation. GFP imaging data were processed using a program sequence setup consisting of an automatic unmixing method with all wavelengths and a manual unmixing method with excitation at 488 nm and emission at 504–770 nm in 20 nm increments for 20 min. The optical fluorescence imaging data were displayed in pseudo-color, representing intensity. Measurements were repeated thrice with different specimens, and the signal intensity of each optical image was calculated within the region of interest.

eGFP fluorescence was visualized at the cellular level using a Zeiss LSM 900 Airyscan 2/GaAsP Confocal Multiplex Microscope (Zeiss, Oberkochen, Germany). Transgenic tissues of Ubi::Cas9–eGFP plants were visualized using a blue filter, which induced green light transmission from the GFP protein.

### 4.10. Measurements of Free Amino Acid Contents

T1 rice seeds of Ramy3D::MdFT1 and ‘Dongjin’ WT plants were ground to a fine powder in liquid nitrogen using a mortar and pestle for free amino acid content analyses. Free amino acid content in the rice seeds was analyzed using a Hitachi LA8080 automatic amino acid analyzer (Hitachi High-Tech Group, Tokyo, Japan). Free amino acid content in rice seeds was measured using the post-column ninhydrin method [66]. Briefly, 50 mg of powdered grain was homogenized with 1 mL of 4% sulfosalicylic acid by vortexing, followed by centrifugation at 1200 rpm for 15 min at 4 °C. Subsequently, the collected supernatant was mixed with an equal volume of 200 mM HCl, then filtrated through a 0.22-µm filter to remove debris. A 20 μL aliquot of filtered sample was injected to measure free amino acid content. Owing to the ninhydrin reaction, a blue–violet substance was produced, and its absorbance was measured at 570 nm.

### 4.11. Data Analysis

Statistical analyses were performed using MS Excel (Microsoft Corp., Redmond, WA, USA) and IBM SPSS (IBM Corp., Armonk, NY, USA). A Student’s *t*-test was used to compare the mean differences in data between pairwise groups (WT and transgenic groups) at the significant level of *p* < 0.05. All data are presented as means ± standard deviations (SD) from multiple replicates of independent experiments.

Calculation Formulae:

Callus induction rate = (the number of induced calli/total numbers of germination seeds) × 100%.

Resistant callus rate = (the number of resistant calli induced/total numbers of callus on selection medium) × 100%.

Genetic transformation rate (transformation efficiency) = (the number of positive transgenic calli/total numbers of resistant calli induced) × 100%.

Shoot regeneration rate = (the number of regenerated shoot/total calli numbers on shoot regeneration medium) × 100%.

## 5. Conclusions

In the present study, we developed an efficient and cost-effective *Agrobacterium*-mediated transformation method for producing transgenic rice plants using seed-derived embryogenic calli in an in vitro tissue culture system. By eliminating the need for complex culture media, we reduced both the time and resources required while maintaining high transformation efficiency. The use of seed-derived embryogenic calli is crucial for consistent and robust regeneration. This method enabled the stable integration and expression of the *eGFP* and *MdFT1* genes in the rice genome. Additionally, the findings of the present study underscore the effectiveness of the *Agrobacterium*-mediated transformation method in gene transfer, particularly when used with strong promoter systems, such as the Ubi and Ramy3D promoters. The introduced traits were inherited stably across generations, thus confirming the reliability of the approach. The process described here is a simplified, scalable method for the genetic transformation of rice, paving the way for enhanced functional gene analysis and the development of genetically modified crops with improved traits.

## Figures and Tables

**Figure 1 plants-13-02803-f001:**
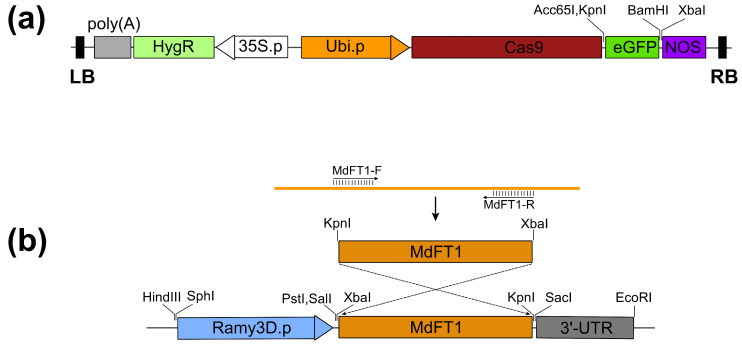
Construction of Ubi::Cas9–eGFP (**a**) and Ramy3D::MdFT1 (**b**) binary vectors for the expression of GOI in rice embryonic calli. The *eGFP* gene was cloned into the binary vector pRGEB32, whereas the *MdFT1* gene was cloned into the pMYD317 expression vector between the XbaI–KpnI sites. LB, T-DNA left border; poly(A), cauliflower mosaic virus (CaMV) polyadenylation signal; *HygR*, hygromycin B phosphotransferase; 35S.p, CaMV 35S promoter; Ubi.p, rice (*Oryza sativa*) polyubiquitin promoter; Cas9, Cas9 (Csn1) endonuclease from the *Streptococcus pyogenes* Type II CRISPR/Cas system; NOS, nopaline synthase terminator; eGFP, enhanced green fluorescent protein; Ramy3D.p, rice alpha-amylase promoter; 3′-UTR, 3′-untranslated region of the rice α-amylase 3D gene; RB, T-DNA right border.

**Figure 2 plants-13-02803-f002:**
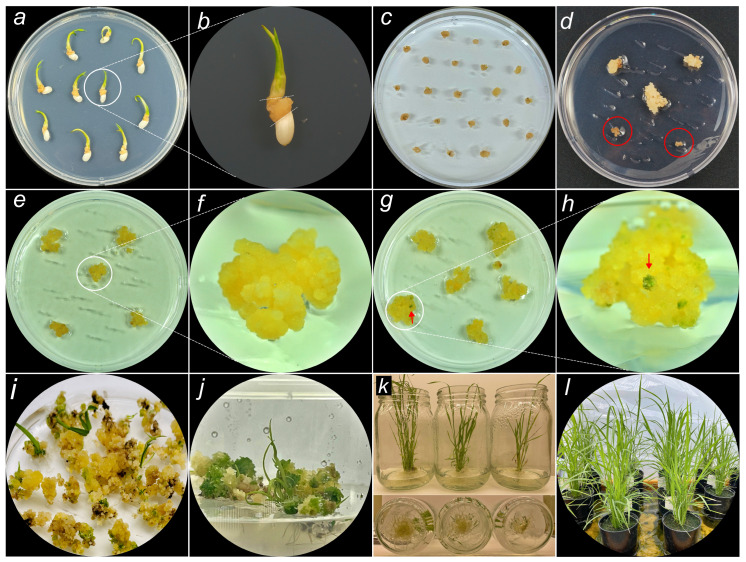
Scheme establishment of transgenic rice using *Agrobacterium*-mediated transformation via in vitro tissue culture. (**a**) Scutellum-derived calli were induced from rice seeds on callus induction medium N6CI. (**b**) Embryonic calli were used as explant for genetic transformation. (**c**) Co-culture of *Agrobacterium* and rice embryonic calli after *Agrobacterium*-mediated transformation. (**d**) Transgenic hygromycin-resistant calli growth on selection medium N6SE. (**e**) Putative transgenic rice calli propagation on N6SE. (**f**) Magnified image of transgenic rice callus line. (**g**,**h**) Growth of transgenic lines on shooting medium MSS with a magnified image. (**i**,**j**) Bud differentiation on MSS after 4 and 6 weeks, respectively. (**k**) Seedling growth on rooting medium MSR. (**l**) Transplantation of fully rooted seedlings for ex vitro culture (sowing into the soil). The red circle in (**d**) indicates that the non-transformed callus cannot grow on N6SE. The red arrow in (**g**,**h**) indicates meristem shoots were initiated from the calli.

**Figure 3 plants-13-02803-f003:**
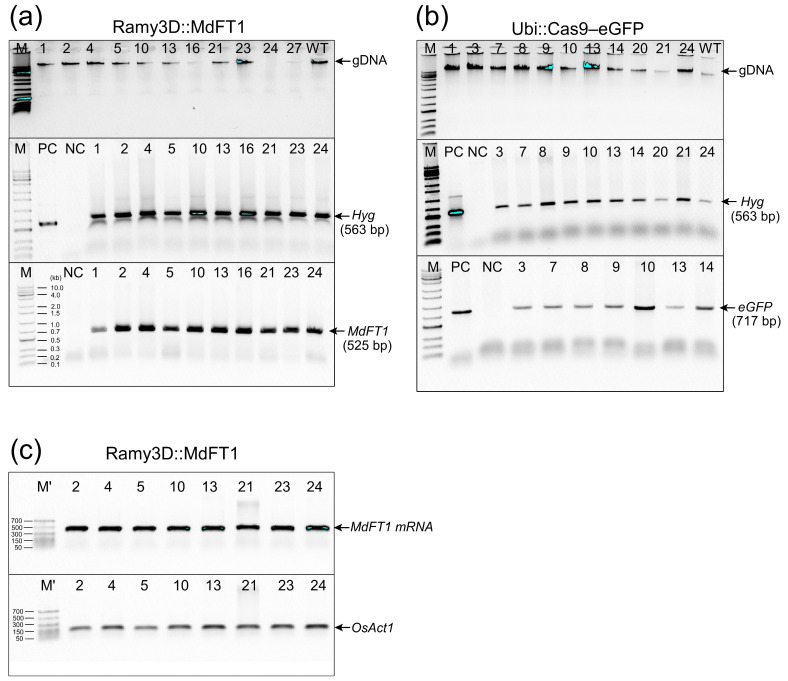
PCR-based screening of transgenic rice calli expressing Ram3D::MdFT1 and Ubi::Cas9–eGFP. (**a**) Detection of the selection marker gene *HygR* and *MdFT1* using specific primers. (**b**) Detection of *HygR* and *eGFP* in Ubi::Cas9–eGFP transgenic lines. (**c**) RT–PCR analysis to evaluate the expression of *MdFT1* mRNA in transgenic rice calli, normalized to the rice reference gene *OsAct1*. M, 1 Kb plus DNA ladder; M’, 50 bp DNA ladder (Dyne Bio Inc. Seongnam-si, Republic of Korea); PC, plasmid DNA of 3D::MdTFL1 and Ubi::Cas9–eGFP (isolated from *E. coli*) as positive control; NC, genomic DNA of WT callus lines as negative control; Numbered lanes, independent transgenic lines; WT, non-transgenic callus genomic DNA.

**Figure 4 plants-13-02803-f004:**
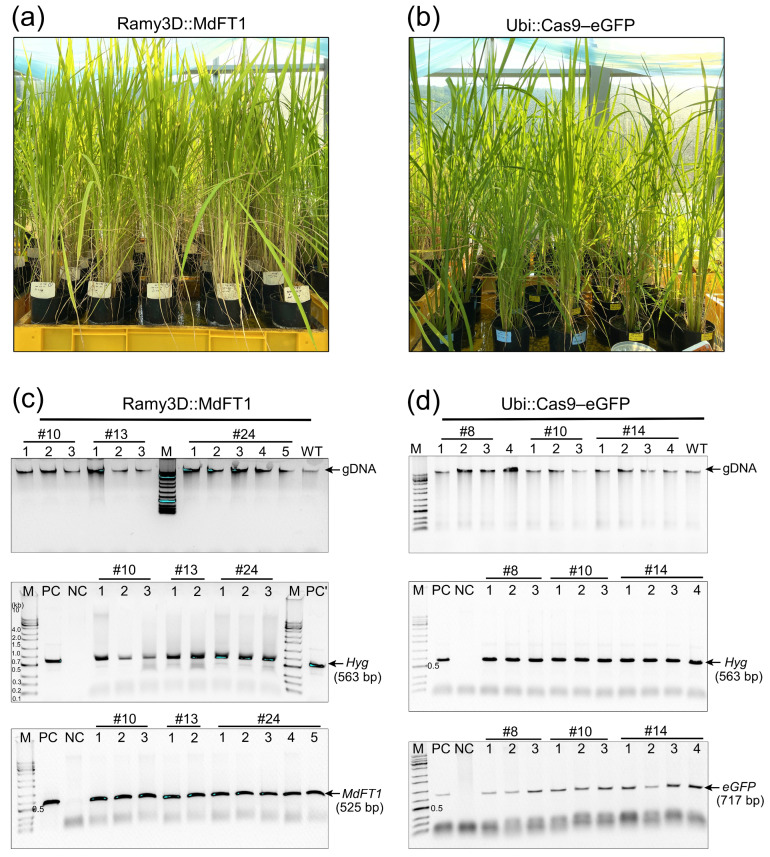
gDNA PCR confirmation of transgenes in T1 transgenic rice plants. Seedlings of transgenic rice Ram3D::MdFT1 (**a**) and Ubi::Cas9–eGFP (**b**) were grown in a greenhouse. Leaves from T1 transgenic plants at the vegetative stage were collected for gDNA PCR. PCR was conducted to detect the *MdFT1* gene (**c**), reporter gene *eGFP* (**d**), and selective marker gene *HypR* (**c**,**d**) using specific primer sets. M, 1 Kb plus DNA ladder; PC, positive control used plasmid DNA of 3D::MdTFL1 and Ubi::Cas9–eGFP (isolated from *E. coli*) as DNA templates; PC’, positive control used genomic DNA of 3D::MdTFL1 (isolated from transgenic callus line No. 1); NC, negative control used genomic DNA of WT lines as DNA templates; Numbered lanes, independent transgenic lines; WT, genomic DNA isolated from leaves of non-transgenic plants.

**Figure 5 plants-13-02803-f005:**
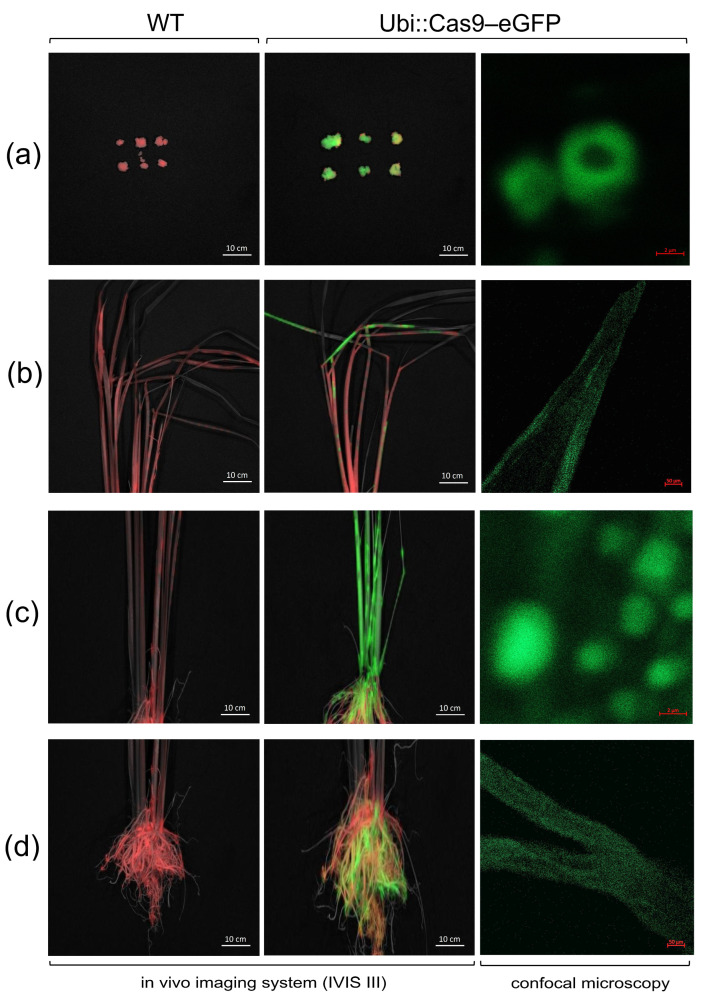
Observation of green fluorescent protein (eGFP) in rice calli (**a**) and various organs of T1 seedlings expressing Ubi::Cas9–eGFP, including leaves (**b**), stems (**c**), and roots (**d**), compared to WT plants. The GFP imaging data were processed using a program sequence that included an automatic unmixing method for all wavelengths, along with a manual unmixing method at 488 nm excitation and emission ranging from 504 to 770 nm, in the IVIS Lumina III system. The cellular eGFP fluorescence in transgenic rice was visualized using a blue filter via the confocal microscopy (described in Section 4.9). The green color is the green fluorescence of GFP and the red color is autofluorescence signals.

**Figure 6 plants-13-02803-f006:**
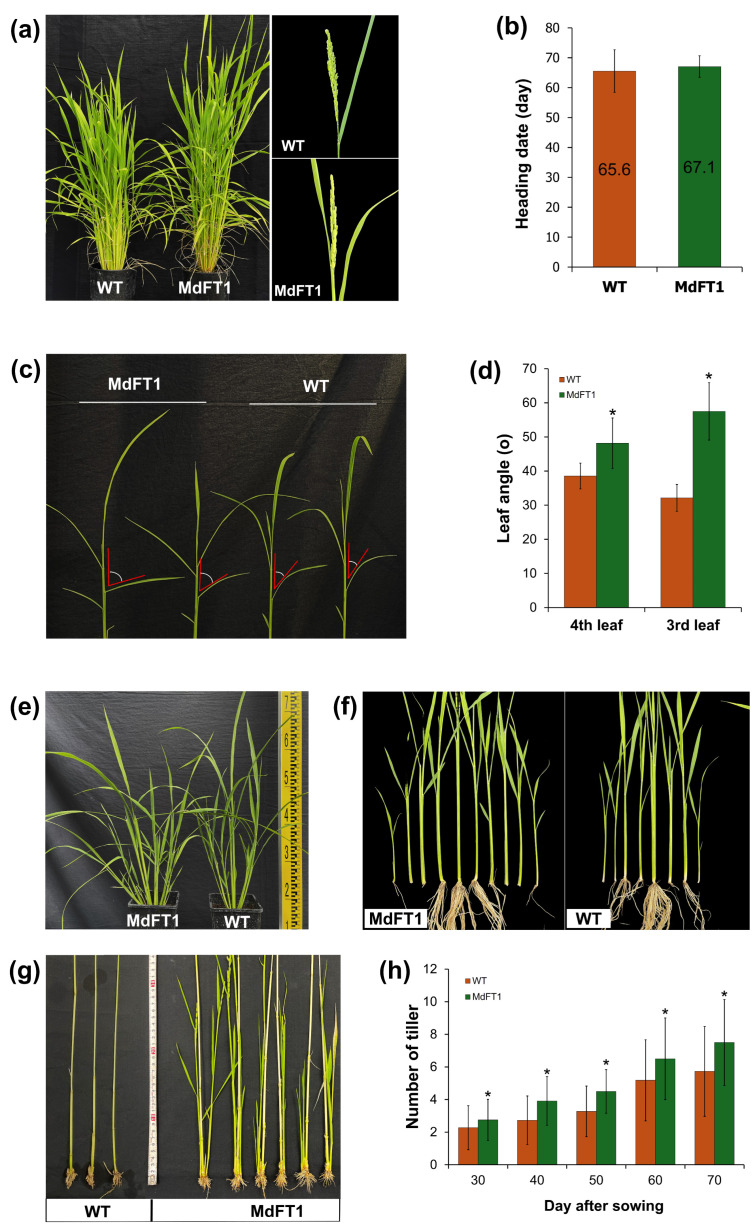
Phenotyping of transgenic rice expressing Ramy3D::MdFT1 compared to WT ‘Dongjin’ cultivar. The analysis includes evaluation of heading dates (**a**,**b**), leaf angle (**c**,**d**), branching at vegetative developmental stage (**e**,**f**), branching at harvest stage (**g**), and the number of tillers during different developmental stages (**h**). Data are presented as means ± standard deviations (SD) (*n* ≥ 20). Asterisks indicate significant differences (*p* < 0.05).

**Figure 7 plants-13-02803-f007:**
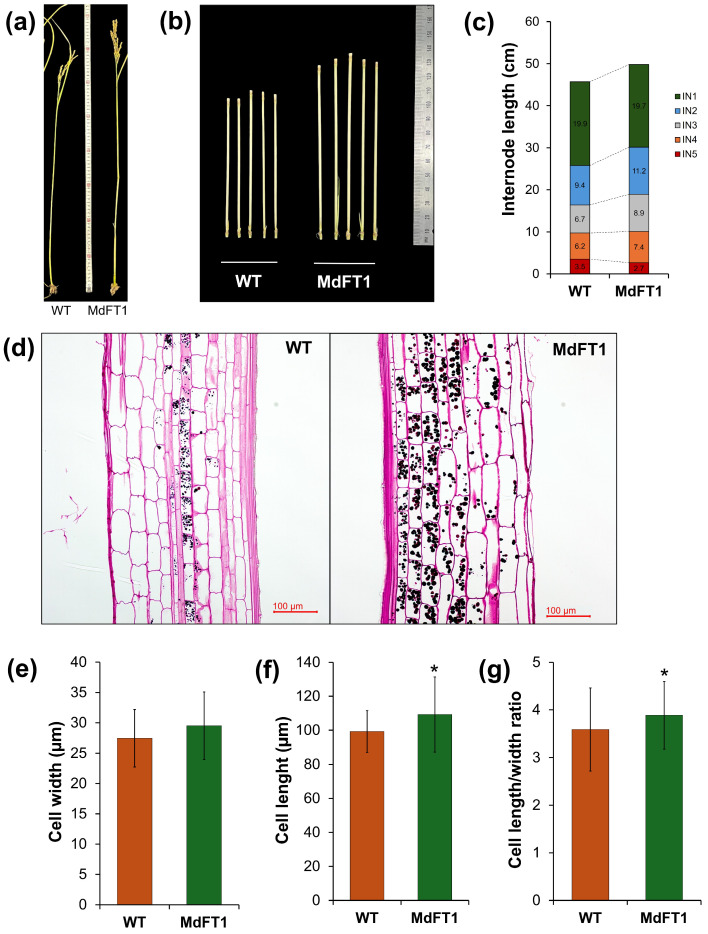
Phenotyping of transgenic rice Ramy3D::MdFT1 in comparison to wild type ‘Dongjin’ cultivar. Transgenic rice developed with taller in overall heights (**a**) and internode (IN2) lengths (**b**,**c**). Microscopy imaging of elongation zone (IN2) (**d**), and cellular size in width (**e**), length (**f**), and length/width ratio (**g**). Data are presented as means ± SD (*n* ≥ 20). Asterisks indicate significant differences (*p* < 0.05).

**Figure 8 plants-13-02803-f008:**
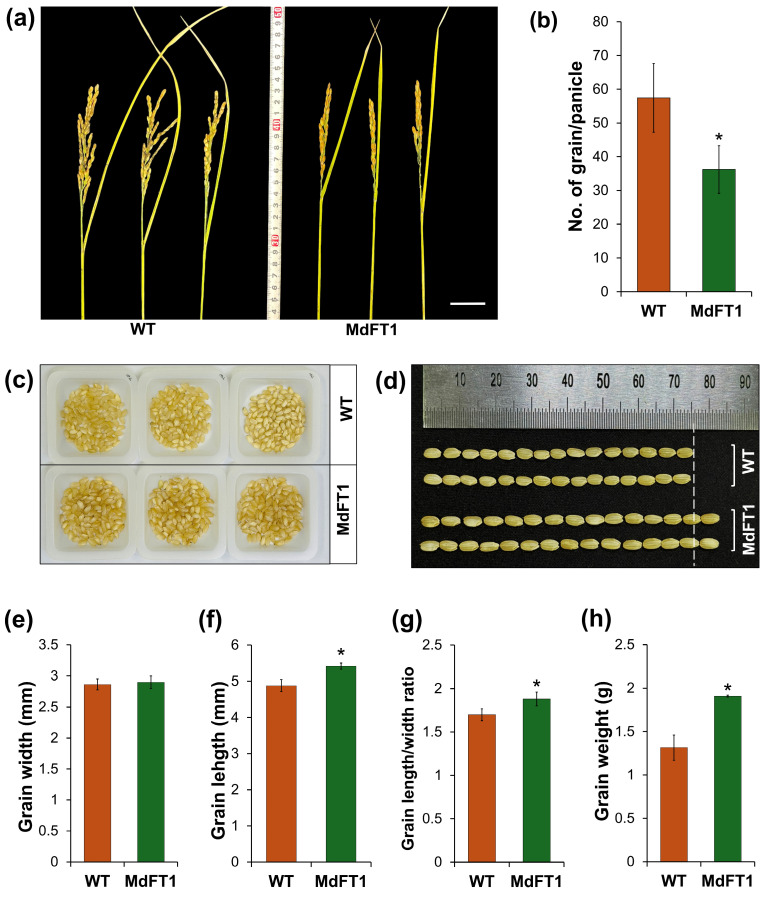
Evaluation of yield potential in transgenic rice expressing *MdFT1*. Rice panicles at ripening stage (**a**) with yield potential (number of grains per panicle) (**b**). Seed production was measured with 100 grains (**c**). Grain size (**d**–**g**) and the average weight of 100 grains (**h**) were also assessed. Data are presented as means ± SD. Asterisks indicate significant differences (*p* < 0.05). Scale bars = 3 cm.

**Figure 9 plants-13-02803-f009:**
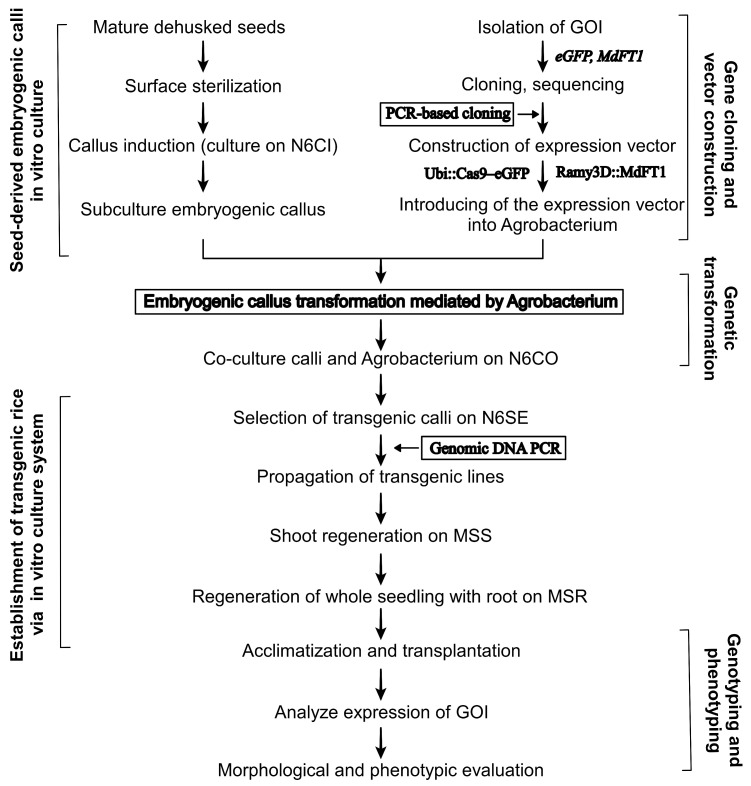
Schematic workflow of the protocol for regeneration of transgenic rice from embryogenic callus via *Agrobacterium*-mediated transformation.

**Table 1 plants-13-02803-t001:** Primer sequences for gene cloning, genomic DNA PCR, and qRT-PCR.

Primer Set	Primer Name		Sequence (5′→3′)	Amplicon Length (bp)	Note
1	*HygR*	F	CTCGGAGGGCGAAGAATCTC	563	Used for gDNA PCR
		R	CAATGACCGCTGTTATGCGG		
2	*eGFP*	F	GTGAGCAAGGGCGAGGAGCT	717	Used for gDNA PCR
		R	TTACTTGTACAGCTCGTCCATGCCGAG		
3	*MdFT1*	F	GGTACC**ATG**CCTAGGGATAGGGAC	525	Used for gene cloning and gDNA PCR and RT–PCR
		R	TCTAGA**TTA**TCTTCTCCTTCCACCG		
4	*OsAct1*	F	GCGTCTGGATTGGTGGTTCT	142	Used for RT–PCR
		R	ACCGCTCTACAAACTTGGCA		

The underlined sequences indicate the restriction enzyme sites. The bold sequences indicate the start and stop codons. F and R denote the forward and reverse primers, respectively.

**Table 2 plants-13-02803-t002:** Transformation efficiency and plant regeneration rate of transgenic rice from embryogenic calli of ‘Samkwang’ and ‘Dongjin’ cultivars.

Construction	Rice Cultivar	Callus Induction (%)	Resistant Calli (%)	Transformation Efficiency (%)	Shoot Regeneration Rate (%)
Ubi::Cas9–eGFP	Samkwang	98.0 ± 2.1	59.4 ± 3.5	53.4 ± 1.6	64.8 ± 8.3
Ramy3D::MdFT1	Dongjin	94.1 ± 3.8	50.1 ± 2.3	47.4 ± 3.4	58.3 ± 8.1

**Table 3 plants-13-02803-t003:** Comparison of free amino acid in grains of WT and Ramy3D::MdFT1 transgenic rice.

Amino Acid (a.a)		Concentration (mg/mL)	
Full Name	Abbreviation	WT	MdFT1
Aspartic acid	Asp	9.04 ± 0.17 ^b^	10.72 ± 0.02 ^a^
Threonine	Thr	3.42 ± 0.08 ^b^	4.13 ± 0.01 ^a^
Serine	Ser	4.03 ± 0.09 ^b^	4.68 ± 0.01 ^a^
Glutamic acid	Glu	15.27 ± 0.32 ^b^	17.38 ± 0.10 ^a^
Glycine	Gly	3.68 ± 0.11 ^b^	4.91 ± 0.03 ^a^
Alanine	Ala	7.16 ± 0.26 ^b^	7.90 ± 0.04 ^a^
Cysteine	Cys	0.88 ± 0.03 ^b^	1.36 ± 0.01 ^a^
Valine	Val	52.68 ± 1.53 ^b^	59.69 ± 0.34 ^a^
Methionine	Met	1.94 ± 0.06 ^b^	2.27 ± 0.01 ^a^
Isoleucine	Ile	3.92 ± 0.12 ^b^	4.48 ± 0.02 ^a^
Leucine	Leu	8.32 ± 0.22 ^b^	9.49 ± 0.02 ^a^
Tyrosine	Tyr	2.19 ± 0.04 ^b^	2.69 ± 0.02 ^a^
Phenylalanine	Phe	5.17 ± 0.11 ^b^	5.97 ± 0.01 ^a^
Lysine	Lys	4.30 ± 0.09 ^b^	5.52 ± 0.01 ^a^
	NH3	1.91 ± 0.10 ^b^	2.02 ± 0.04 ^a^
Histidine	His	2.31 ± 0.05 ^b^	2.82 ± 0.00 ^a^
Arginine	Arg	7.08 ± 0.15 ^b^	7.80 ± 0.01 ^a^
Proline	Pro	3.63 ± 0.72 ^b^	4.66 ± 0.93 ^a^

Data are means ± SD (*n* = 2). Means followed by different letters within columns of concentration represent significant differences between WT and transgenic MdFT1 using a *t*-test (*p* < 0.05).

**Table 4 plants-13-02803-t004:** Composition of the media used for plant tissue in vitro culture.

Component	Producer /Product No.	Callus Induction (N6CI)	Co-Culture (N6CO)	Selection (N6SE)	Shooting (MSS)	Rooting (MSR)
CHU (N6) plus vitamins	Duchefa Biochemie C0204.0050	4.0	4.0	4.0		
MS plus vitamins	Duchefa Biochemie M0222.0050				4.3	4.3
Sucrose	Duchefa Biochemie S0809.100	30	30	30	30	30
Glucose	Sigma Aldrich G5767-500G		10			
Sorbitol	Sigma Aldrich S3889-1KG				30	
Phytagel	Sigma Aldrich P8169-500G	2.3	2.3	2.3		
Gerlite	Duchefa Biochemie G1101.0500				4.0	4.0
2,4-D	Duchefa Biochemie D0911.0250	2 mg/L	2 mg/L	2 mg/L		
Kinetin	Duchefa Biochemie K0905.005	2 mg/L	2 mg/L	2 mg/L	5 mg/L	
NAA	Duchefa Biochemie N0903.0025				1 mg/L	
IBA	Sigma Aldrich I5386-5G					0.5 mg/L
Acetosyringone	Sigma Aldrich D134406-5G		100 µM			
Cefotaxime	Duchefa Biochemie C0111.0005			250 mg/L	250 mg/L (OP) *	250 mg/L (OP)
Hygromycin B	Duchefa Biochemie H0192.0001			50 mg/L	50 mg/L (OP)	50 mg/L (OP)
pH		5.8	5.2	5.8	5.8	5.8

All hormones (2,4-D, Kinetin, NAA, IBA), antibiotics (Cefotaxime, Hygromycin B), and acetosyringone were of analytical grade. They were prepared in stock solutions (1.0 mg/mL for all hormones); 100 mM for acetosyringone; 100 mg/mL for Cefotaxime; and 50 mg/mL for Hygromycin B, and then sterilized using a 0.2-µm filter. * OP indicates optional, because transgenic lines had already been confirmed by gDNA PCR at the selection stage of in vitro cultivation.

## Data Availability

Data are presented within the article and Appendix A.

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
