# Peer review of "Efficient Regeneration of Transgenic Rice from Embryogenic Callus via Agrobacterium-Mediated Transformation: A Case Study Using GFP and Apple MdFT1 Genes"

_plants, 2024, doi:10.3390/plants13192803_

Round 1
Reviewer 1 Report
Comments and Suggestions for Authors
Constructing a plant genetic transformation method is complex and time-consuming. The successful integration of foreign genes into the genomes of two Korean cultivars, as well as the detailed information of the method described in this manuscript will provide an important technical reference for the development of rice genetic transformation system. However, I have some confusions and suggestions about this manuscript here.
Title
In the title, you state that the method described in your manuscript is an efficient genetic transformation system for rice. You also use the word "efficient" several times throughout the paper. However, there is no statistical data provided to support the claimed efficiency of regeneration and transformation. For instance, how many transgenic plants were generated? How many starting explants were used to obtain these transgenic plants? Additionally, how many plants can a single transgenic callus produce?
Introduction
The background information on rice genetic transformation in the introduction part is insufficient. Agrobacterium-mediated genetic transformation has been established for several rice varieties for decades. The difficulty and efficiency of genetic transformation vary significantly between indica and japonica rice. Therefore, what is the current research status of rice genetic transformation, particularly, for the cultivars you used in your study, and why did you choose these two cultivars as your recipient materials? Addressing these points will help highlight the value and significance of your research.
Results
The pictures in Figure 1, especially the screenshot of the plasmid map, are blurry and do not meet the required standards.
In your paper, you used two binary expression vectors with different promoter systems, Ubi and Ramy3D. I wonder if you have compared the gene expression levels under the different promoter systems to evaluate the promote efficiency of the two different promoters.
Line 120, what is the exact number of the transgenic lines for the two vectors you obtained? If possible, please specify it in the text.
Line 122-123, the sentence: “seed‒derived embryogenic calli were used as explants for genetic transformation using Agrobacterium‒mediated transformation”, could be changed to “seed‒derived embryogenic calli were used as explants for Agrobacterium mediated genetic transformation” to avoid repetition
Line 159-160, it may be better to replace “Ram3D::MdFT1” and “Ubi::Cas9‒eGFP” with the vector names “pRGE32-eGFP”and“pMYD317”, as HygR gene is on the vector and not included in the GOI fragment region.
For Figure 3, I don’t think you should include the picture of the DNA ladder (Fig 3d), as it is not part of your results. Simply marking it on your result picture to indicate the target gene band would be enough. Additionally, the image quality is unclear, making it difficult to see detailed information.
Line 230-232, it should be better to revise the sentences as “Transgenic Ramy3D::MdFT1 rice showed a slight delay in heading date (HD) compared to the wild type (WT), with an average HD of 65.6 days in WT and 67.1 days in Ramy3D::MdFT1, although the differences were not statistically significant (Figure 6b)”.
Discussion
Line 316, “recipient materials” should be better than “explant materials”.

Comments on the Quality of English LanguageOverall, the English in this manuscript is readable and easy to understand except few sentences, which are repetitive, complicated and lack clarity. I recommend a thorough review for grammar errors, repeated word, and sentence structure to enhance the readability, clarity and precision of this paper.
Reviewer 2 Report
Comments and Suggestions for Authors
The manuscript «Efficient Regeneration of Transgenic Rice from Embryogenic Callus via Agrobacterium-Mediated Transformation: A Case Study using GFP and Apple MdFT1 Genes Van Giap Do, Seonae Kim, Nay Myo Win, Soon-Il Kwon, Hunjoong Kweon, Sangjin Yang, Juhyeon Park, Gyungran Do and Youngsuk Lee is a study examining Agrobacterium-mediated transformation of rice. The topic is undoubtedly relevant, but the manuscript requires significant improvement.
Introduction.
Unfortunately, the introduction provides very general information on the transformation. There is virtually no information on previous work on rice transformation, including with the help of Agrobacterium, as well as the bioballistics method. What are the known successes and problems in rice transformation?
There is no real information on the advantages of Agrobacterium-mediated transformation, given that the "gold standard" in the transformation of monocots is bioballistics.
What is the rationale for choosing the genes of interest, vectors, and agrobacterium strain?
I believe that this information should be added to the introduction, removing the general information about the transformation as a whole that is well known to everyone.
Results.
The main question is: what was the efficiency of the transformation and regeneration of plants?
I believe that the vectors in Figure 1 should be presented in a single style.
The expression "Meristem shoots" (line 131) is unsuccessful. It is better to replace it with "Bud differentiation" line 147 (as in the caption to Figure 2).
Figures 3 and 4 show the PCR results. What do the numerous tracks on the gels mean? Different rice lines? They need to be labeled. Why are there so many of them?
Figure 5. The caption to the figure should include the parameters for obtaining images. Fluorescence excitation length and fluorescence wavelength range. What does the red signal mean and how is it obtained. The images obtained on the confocal microscope are extremely fuzzy. It is imperative to add a scale bar. What is shown here?
The article mentions two rice varieties, ‘Samkwang’ and ‘Dongjin’. Were both varieties transformed? For which variety are the results shown? Were there any differences?
The authors note in their results that they observed a decrease in the time of the onset of heading of the transgenic rice compared to the original. And they refer to Figure 6. No differences are shown there. Correct the text.
Figure 7d should show photographs of the internodal cells on the same scale. Currently, the scale is different, which misleads readers.
Did the insertion of the eGFP gene affect the parameters shown in Figures 6 and 7?
A scale bar should be provided in Figure 8a.
Which results in Table 2 were processed using ANOVA? Show.
Discussion.
Unfortunately, this section does not discuss the results or compare them with previous works.
The authors talk about the better efficiency of the method they developed. However, the results do not contain this information, and the discussion does not compare with other works.
It is not sufficiently substantiated why the MdFT1 protein gene was used. It is necessary to give a brief description of this protein and its role at least here or in the introduction. Have there been similar studies on the transformation of other plants with this protein gene?
Methodology.
What is the N6CI medium (line 436)? Who developed it? Is it liquid? How many ml and on what dish was it poured? Were there any growth regulators in this medium? Why is this medium considered simpler? Compared to what media?
Please write point 4.4 more clearly. In my opinion, the information in this point is duplicated.
Under what conditions (lighting, temperature) were calli and regenerants cultivated in vitro?
Point 4.11. Data analysis. What statistical criterion was used?
Round 2
Reviewer 2 Report
Comments and Suggestions for Authors
The authors have done a lot of work to improve the article.
Unfortunately, the introduction has become longer, but not more informative. General information about the types of transformation has been added. I think that the text should not be overloaded with general phrases about plant transformation and dicotyledons should not be mentioned so widely. I suggest focusing on monocotyledons. There is still no information about previous work on rice transformation. I would like to note that there are many such works, so it is necessary to clearly justify the novelty of this work.
The caption to Figure 5 is still not informative. Prove that you received a GFP signal. What does the red color mean in the images? Add a scale bar. Especially in confocal microscope images.
Edit the font on the scale bar in Figure 7d .
